# Frontal Lobe Functions, Demoralization, Depression and Craving as Prognostic Factors and Positive Outcomes of Patients with Heroin Use Disorder Receiving 6 Months of Methadone Maintenance Treatment

**DOI:** 10.3390/ijerph19063703

**Published:** 2022-03-20

**Authors:** Ting-Gang Chang, Ting-Ting Yen, Wen-Yu Hsu, Shan-Mei Chang

**Affiliations:** 1Department of Psychiatry, Taichung Veterans General Hospital, Taichung 407219, Taiwan; ctg1013@vghtc.gov.tw; 2School of Psychology, Chung Shan Medical University, Taichung 402306, Taiwan; 3Department of Otorhinolaryngology, Taichung Veterans General Hospital, Taichung 407219, Taiwan; tingting@vghtc.gov.tw; 4School of Medicine, National Yang-Ming Chiao-Tung University, Taipei 112304, Taiwan; 5Department of Psychiatry, Changhua Christian Hospital, Changhua 500209, Taiwan; 117006@cch.org.tw; 6School of Medicine, Chung Shan Medical University, Taichung 402306, Taiwan; 7Graduate Institute of Clinical Medical Science, China Medical University, Taichung 406040, Taiwan; 8School of Nursing, China Medical University, Taichung 406040, Taiwan; 9Nursing Department, China Medical University Hospital, Taichung 404332, Taiwan

**Keywords:** methadone, craving, depression, frontal assessment battery, demoralization

## Abstract

Methadone maintenance therapy (MMT) is a well-established and effective treatment for heroin use disorders. Whether frontal lobe function and demoralization serve as suitable prognostic and outcome assessment factors remains unknown. A quasi-experimental study was conducted with a single-group repeated-measures design at a medical center and mental hospital in Taiwan. We enrolled 70 participants (39 completed treatments and 31 dropped out). Frontal lobe function, demoralization, depression, and craving at three time points were analyzed. There were differences between patients who completed the treatment (n = 39) and those who did not (n = 31). Thirty-nine patients completed the treatment (average age, 45.5 years; 89.7% men; average duration of heroin use, 27.21 years; MMT, 38.18 mg/day). Post-MMT (6 months), frontal lobe function, demoralization, depression, and craving significantly improved. Dropouts had higher frontal lobe function, lower demoralization, higher craving, younger age, and earlier onset age than patients who completed the pretest treatment. Clinicians should be aware of the severity of demoralization. Clinicians may select suitable patients for MMT by assessing frontal lobe function, demoralization, craving, age, and onset age. A 6-month course of MMT improved demoralization, frontal lobe function, depression, and addiction. Six months of treatment was more effective than 3 months. Suitable patient identification and continuous treatment are important in MMT.

## 1. Introduction

Methadone maintenance treatment (MMT) is the most widely used harm reduction approach for the treatment of opioid use disorders. Methadone is a safe, low-cost, and convenient generic drug used for the treatment of opioid dependence [1]. Various aspects of life are significantly improved by MMT, such as health, social functioning, quality of life, and other positive outcomes [2]. Age, marital status, employment status, sex, and duration of treatment have been shown to influence treatment outcomes and may be valuable prognostic factors in patients receiving MMT [3,4]. Previous studies have shown the significant effectivity of such treatment on the psychopathological background of addiction including depression and anxiety [5,6]. Nevertheless, there have been few studies on frontal lobe function and demoralization in patients undergoing MMT.

### 1.1. Brain Frontal Lobe Function

The frontal lobe function is impaired as a result of long-term substance use [7,8,9,10]. The frontal cortical areas of the brain oversee behavioral control through executive functions. Executive functions include abstract thinking, motivation, planning, attention to tasks, and the inhibition of impulsive responses. Abstinence from substance use results in bursts of neurogenesis and brain regrowth [11]. Executive function problems may be a risk factor for dropout and subsequent relapses in substance users [12,13]. Moreover, Rezapour et al. stated that baseline executive function can predict the outcome of 3 months of MMT [14]. The positive and negative effects of MMT on frontal lobe dysfunction have not been confirmed, and studies on frontal lobe function as a prognostic factor and treatment outcome are therefore needed.

### 1.2. Addiction and Demoralization

Demoralization is a phenomenon of existential suffering and loss of meaning in life, which differs from the diagnostic criteria of major depressive disorder according to the American Psychiatric Association. Demoralization may be a normal psychological response to painful, advanced, and/or terminal diseases [15,16] and currently does not refer to specific brain pathology. In contrast to major depression, demoralization usually manifests as existential distress, helplessness, hopelessness, and the loss of meaning and purpose associated with a specific event [17]. When patients are distracted by thinking about events or situations, they usually feel normal. In clinical situations, many patients with chronic and severe medical diseases are not diagnosed with major depression but desire to die because of loss of meaning and purpose [18]. Demoralization frequently occurs in patients with opioid-use disorder [19]. Moreover, a recent study revealed that individuals recovering from opioid use disorder may suffer from demoralization, which is related to prefrontal cortex function [20]. Demoralization and opioid abuse have been identified as important risk factors for suicide [21]. Huhn et al. also suggested that demoralization is an important outcome of MMT.

### 1.3. The Present Study

Therefore, the purpose of this study was:To explore the severity of demoralization in patients receiving MMT.To explore demoralization or frontal lobe function as prognostic factors to predict retention rate.To explore whether individuals with heroin use disorders who received MMT showed improvement in frontal lobe function, demoralization, depression, and craving after treatment.

## 2. Materials and Methods

### 2.1. Design

Longitudinal panel data were analyzed to evaluate participants receiving MMT at three time points (initial visit, treatment for 3 months, and treatment for 6 months) concerning the treatment outcomes of therapy, including brain frontal lobe function, demoralization, depression, and craving. Risky alcohol use refers to the definition of the National Institute on Alcohol Abuse and Alcoholism in the United States, and current cigarette smoking refers to those who smoked at the time of enumeration. Past illicit drug use refers to the past use of illicit drugs in Taiwan, including cannabis, methamphetamine, MDMA, and GHB, other than opioids, before receiving this treatment. This study was conducted in the methadone clinic of a medical center. Treatments in addition to MMT, such as psychosocial interventions and pharmacological treatment, were not provided in the study.

### 2.2. Participants

A total of 70 participants were enrolled (39 completed the treatment course and 31 partially completed the treatment course). Patients were included according to the following criteria: (1) diagnosed with opioid use disorder according to the DSM-5 criteria, (2) aged 18 to 65 years, and (3) willing to accept methadone replacement therapy for at least 24 weeks. The non-inclusion criteria were as follows: (1) unwillingness to accept alternative methadone therapy; (2) having acute or serious physiological problems that may prevent the completion of the study; and (3) having received any treatment for opioid addiction (including methadone maintenance therapy) within the preceding 30 days.

### 2.3. Instruments

#### 2.3.1. Demoralization (DS-MV)

DS-MV is a demoralization scale developed by Kissane [22]. The DS-MV is composed of five distinct dimensions: loss of meaning (five items), disheartenment (six items), dysphoria (five items), sense of failure (four items), and helplessness (four items). The Mandarin version for patients with cancer had a Cronbach’s alpha of 0.92 [23]. According to previous studies, a DS-MV score higher than 30 points denotes high demoralization [22,23].

#### 2.3.2. The Chinese Craving Scale (CCS)

The Chinese Craving Scale (CCS) consists of 10 items rated on a 4-point scale, adapted from the Craving Belief Questionnaire (Beck, 1993). Higher scores indicate stronger cravings. The CCS has acceptable internal consistency (Cronbach’s α = 0.90) based on a large sample size study (n = 958) performed in Taiwan [24].

#### 2.3.3. The Patient Health Questionnaire (PHQ-9)

The PHQ-9 is a 9-question instrument given to patients in a primary care setting to screen for the presence and severity of depression. The results of the PHQ-9 can be used to establish a diagnosis of depression according to the DSM-5 criteria. Test-retest reliability was assessed by the correlation between the PHQ-9 scores obtained from in-person and phone interviews with the same patients. The obtained correlation value was 0.84 [25]. In an assessment of construct validity, the correlation coefficient between the PHQ-9 and SF-20 mental health scales was 0.73. To assess criterion validity, a mental health professional validated depression diagnoses using PHQ-9 scores from 580 participants, resulting in 88% sensitivity and 88% specificity [25].

#### 2.3.4. Taiwanese Version of Frontal Assessment Battery (TFAB)

The frontal assessment battery (FAB) was developed as a short-bedside cognitive and behavioral battery to assess frontal lobe function. The test-retest reliability and criterion-related validity of the Taiwanese version of the Frontal Assessment Battery (TFAB) (Wang et al., 2016) are good (Wang et al., 2016). The TFAB consists of six items, and the score for each item ranges from 0 to 3, with higher scores indicating better performance. The six subtests of the FAB explore (1) similarities (conceptualization), (2) lexical fluency (mental flexibility), (3) Luria motor sequences (programming), (4) conflicting instructions (sensitivity to interference), (5) a go/no-go test (inhibitory control), and (6) prehension behavior (environmental autonomy).

### 2.4. Ethics

This study complied with the latest version of the Declaration of Helsinki. The Changhua Christian Hospital Clinical Research Ethics Committee approved the study (CCH-IRB-No: 181120) and informed consent was obtained from all participants.

## 3. Results

### 3.1. Characteristics of the Participants

Seventy individuals participated in this study. Thirty-nine subjects completed the study, and 31 did not complete the study. There was no significant difference in the relevant background information between the intervention and dropout groups (Table 1).

Table 1 and Table 2 include the descriptive characteristics of all participants, who were divided into two groups: the fully attended group and the dropout group. More than 88% of the participants were male (n = 62, 88.6%), and their age ranged from 34 to 71 years (M = 43.79, SD = 8.23). The mean age of onset of heroin use disorder was 25.49 years (SD = 9.33), and the average dose of MMT was 38.18 mg/day (SD = 9.33). The majority of the subjects had only graduated from junior high school (n = 39, 55.7%), followed by senior high school (n = 22, 31.4%), primary school (n = 6, 8.6%), and university/college (n = 3, 4.3%). Married participants accounted for the largest proportion (n = 21, 30%), followed by single (n = 21, 28.6%), divorced (n = 18, 25.7%), cohabiting (n = 3, 4.3%), and deceased (n = 1, 1.4%) participants. More than half were employed (n = 42, 60.0%), followed by the unemployed (n = 21, 30%), and underemployed (n = 7, 10%). Ten subjects were HIV-positive (14.3%), 70 were current cigarette smokers (100%), 19 were risky alcohol users (27.1%), and 68 were past illicit drug user (97.1%). The *p*-values of the fully attended group (n = 39) and dropout group (n = 31) were not significant (see Table 1).

Concerning the 70 participants’ attendance rate, 15 (21.4%) had a rate of less than 70%, 17 (24.29%) had a rate of 70%–89%, 28 (40%) had a rate of 90%-99%, and 10 (14.28%) had full attendance (see Table 1).

The mean differences were analyzed among the average methadone doses, and the 1st DS-MV, PHQ-9, CS, and TFAB scores of the 70 participants were studied. The Kolmogorov–Smirnov test was used to test the normality of data distribution, and the results indicated that the age, onset age, 1st TFAB, and 1st CS total scores did not conform to a normal distribution. Variables with a normal distribution were analyzed using the independent sample t-test, and those without a normal distribution were analyzed using the Wilcoxon test. The results revealed that age, onset age, and scores on the 1st TFAB, 1st DS-MV, and 1st CS were significantly different between the full attendance and dropout groups (age: w = −2.49, *p* = 0.013 *; onset age: w = −2.17, *p* = 0.030 *; 1st TFAB: w = −2.03, *p* = 0.043 *; 1st DS-MV: t = −2.18, *p* = 0.033 *; 1st CS total scores: w = −2.09, *p* = 0.037 *). However, the average dose of methadone (mg/day) and scores of the 1st PHQ-9 depression scale were not significantly different between the full attendance and dropout groups (see Table 2).

### 3.2. Treatment Outcomes

One-way repeated-measures ANOVA showed significant differences in frontal lobe function between the three time points (TFAB total scores), F _(2, 74)_ = 7.662, *p* = 0.001. Pairwise comparisons revealed that the average total of the 1st TFAB (M = 36.21, SD = 12.42) was significantly lower than that recorded in the 3rd TFAB (M = 39.92, SD = 10.94) (*p* = 0.005). In addition, the 2nd TFAB (M = 35.97, SD = 12.00) was significantly lower than that recorded in the 3rd TFAB (*p* < 0.001); there was no difference between the average total intake recorded in the 1st TFAB and 2nd TFAB (*p* = 0.809). This indicated that frontal lobe function (TFAB scale score) significantly improved following treatment for six months (Table 3).

One-way repeated-measures ANOVA showed significant differences between the three time points for demoralization (DS-MV total scores), F _(2, 76)_ = 3.435, *p* = 0.037. Pairwise comparisons revealed that the average total of the 1st DS-MV (M = 36.49, SD = 14.19) was significantly higher than that recorded in the 3rd DS-MV (M = 29.92, SD = 15.13) (*p* = 0.011), but the 2nd DS-MV (M = 32.69, SD = 17.06) and 3rd DS-MV (*p* = 0.176) showed no differences; there was no difference in average total intake between the 1st DS-MV and 2nd DS-MV (*p* = 0.213). This indicated that demoralization (DS-MV scale score) significantly improved following continuous treatment for six months (Table 3).

One-way repeated-measures ANOVA showed significant differences among the three time points for depression (PHQ-9 total scores), F _(1.568, 59.587)_ = 10.166, *p* < 0.001. Pairwise comparisons revealed that the average total score on the 1st PHQ-9 (M = 8.62, SD = 6.97) was significantly higher than that recorded on the 2nd PHQ-9 (M = 5.97, SD = 5.92) (*p* = 0.032), and the score on the 2nd PHQ-9 was significantly higher than that recorded on the 3rd PHQ-9 (M = 4.28, SD = 4.26) (*p* = 0.034). This indicates that depression (PHQ-9 scale score) significantly improved following continuous treatment for six months (see Table 3).

One-way repeated-measures ANOVA showed significant differences among the three time points for craving symptoms (CS total scores), F _(1.64, 62.16)_ = 9.51, *p* < 0.001. Pairwise comparisons revealed that the average total of the 1st CS (M = 9.79, SD = 6.89) was significantly higher than that recorded in the 3rd CS (M = 4.54, SD = 5.33) (*p* < 0.001), the 1st CS was significantly higher than that recorded in the 2nd CS (M = 6.13, SD = 7.52) (*p* = 0.015), and there was no difference in the average total intake between the 2nd CS and 3rd CS (*p* = 0.093). This indicates that craving (CS score) significantly improved after the first three months of treatment (see Table 3).

## 4. Discussion

This study had four main findings: 1. In the present study, a DS-MV score higher than 30 points was defined as high demoralization, and the severity of demoralization was high in patients receiving MMT. 2. Low frontal lobe function and high demoralization, early-onset age, and severe craving may be used as predictors of the retention rate of MMT. 3. MMT provided overall positive effects on frontal lobe function after six months of MMT. 4. The outcomes of MMT, including demoralization, frontal lobe function, depression, and craving, were better at six months than at three months.

It is worth noting that patients with a worse frontal assessment battery were more likely to continue receiving MMT. This may be because participants in the dropout group were relatively young and had early onset, which might have better preserved the frontal lobe function. Another possible reason is that methadone is a low-cost and convenient generic drug for the treatment of opioid dependence. Individuals with a better frontal assessment battery may be able to find alternative ways to deal with heroin use disorders. In addition, in the community, the use of methadone is stigmatized as it is considered an addictive drug; therefore, those who receive MMT are deemed to be “addicts” who should be avoided [26]. In the present study, the dose of methadone used was relatively low. The reason may be the personalization of the dose [27] or the patient being heisted or refusing to increase the dose. In addition to methadone treatment, patients may seek alternative treatments with less structure and greater autonomy [28]. Many studies have reported that higher and more stable doses are related to treatment effectiveness and retention [3]. However, there is considerable evidence, such as the results of a driving-relevant psychomotor battery, that methadone has possible side effects [29]. Alternative treatments, such as buprenorphine, may explain why patients with a higher frontal assessment battery score withdrew from treatment. Nevertheless, clinicians should recommend methadone or buprenorphine-naloxone for abstinence-based treatment [30].

The present study found that the frontal assessment battery scores of patients with heroin use disorder were worse than those of healthy individuals, which is consistent with previous studies showing that patients with heroin use disorder had poor frontal lobe function and that chronic heroin users had cumulative impairment of frontal function [31,32]. Several studies have found that chronic MMT treatment may worsen some executive functions [29]. New research has found that only impaired associative learning related to depressive and anxiety symptoms may be impaired [33]. This study found that although MMT may have an adverse effect on cognitive function, frontal lobe function gradually improved after 6 months of MMT. Volkow (2011) stated that a combined biological treatment and cognitive function intervention can normalize prefrontal function in patients with addiction [31]. Although the initial treatment may not be effective, the improvement of frontal lobe function can be seen when the 6-month treatment is completed.

This investigation is one of the few studies to explore the demoralization of patients with heroin use disorders before participating in MMT. The present study found that the mean DS-MV score of all participants was 39.73, indicating high demoralization. The severity and prevalence of depression and demoralization are not necessarily related, and the correlation between depression and demoralization is similar in patients with cancers [34,35]. The demoralization scores of participants were higher in the dropout group. Highly demoralized patients may feel more hopeless and, hence, more likely to withdraw from MMT. Therefore, clinical staff must be reminded that depression and demoralization have different symptoms, and that demoralization may confer a higher risk of suicide [36]. Demoralization must be properly evaluated, and a treatment model must be involved [37]. The present study provides evidence that MMT can improve demoralization, although the underlying mechanisms remain unclear.

The severity of depression in patients decreased with longer MMT durations. In the literature, it was found that comorbid depression and heroin use disorder were associated with a lower retention rate, and women were more likely to have comorbid depression than men [38]. The severity of depression in this study was not significantly different between the two groups and was less than that of mild depression. This finding may be related to the sex ratio distribution, which was predominantly male.

In the first 3 months, MMT significantly improved craving symptoms. Therefore, it is unfortunate that patients withdrew from treatment within 3 months. Craving is significantly associated with the level of heroin use disorder and MMT adherence [39]. Previous studies have found that if patients experience more psychiatric symptoms, they may more readily withdraw from treatment, but if they receive treatment for a longer time, the improvement will be more significant [40,41]. Long-term MMT may improve the heroin-craving response by modulating impaired function in the bilateral dorsal striatum caused by heroin use [42]. Clinicians need to remind patients that craving can be significantly improved in the first 3 months and that craving will continue to decline after continuous treatment.

Based on the results of our study, we recommend the following clinical implications: Identifying patients with high frontal lobe function, providing a more autonomous treatment model, and destigmatizing the use of methadone are important issues. Rating scales for depression may not be able to detect demoralization. In clinical practice, DS-MV may be used to evaluate motivation for MMT. Reducing demoralization and enhancing the motivation to participate in MMT may be directions for future research.

### 4.1. Strengths

The main strength of the study is that it was a 6-month prospective study and the first to investigate demoralization as a prognostic factor in patients receiving MMT.

### 4.2. Limitations

This was a preliminary study. The participants were from a single institution, and the number of cases was relatively small. The present study did not provide reports of urine opioid screen tests and heroin use status during MMT. The mean dose of methadone in participants was low and the dispersion was large, and the negative effects of methadone on frontal lobe function might have been hidden. In general, treatment outcomes are based on the global care of MMT programs, and it is difficult to distinguish the difference between the pharmacological effect of methadone and the effect of stabilization and global care. 

Our assessment of frontal lobe function only provided an overall percentile rating and did not provide sub-items of function. The present study only recorded the status of tobacco and alcohol use at the initial stage but did not record the use of alcohol and cigarettes during the treatment process. We used the threshold of a DS-MV score of 30 points as high demoralization and did not compare with the control group, for example, the healthy population. This study did not provide patients’ past treatment records or efficacies for opioid use disorders. 

## 5. Conclusions

The severity of demoralization was high in the patients receiving MMT. Low demoralization, high low lobe function, younger age, and early onset age may predict MMT retention. Six months of MMT significantly improved frontal lobe function, demoralization, depression, and craving. Clinicians should identify patient needs early and suggest appropriate treatment models. This was a preliminary study, and larger studies are required.

## Figures and Tables

**Table 1 ijerph-19-03703-t001:** Descriptive characteristics of all participants, full attendance group, and dropout group.

Variable		All Participants n = 70		Full Attendance Group n = 39		Dropout Group n = 31		*p*
	Items	Number of People	%	Number of People	%	Number of People	%	
gender	male	62	88.6	35	89.7	27	87.1	0.51
	female	8	11.4	4	10.3	4	12.9	
education	primary school	6	8.6	4	10.3	2	6.5	0.67
	junior high school	39	55.7	23	59.0	16	51.6	
	senior high school	22	31.4	10	25.6	12	38.7	
	university/college	3	4.3	2	5.1	1	3.2	
marital status	single	27	28.6	14	35.9	13	41.9	0.46
	married	21	30.0	12	30.8	9	29	
	divorced	18	25.7	9	23.1	9	29	
	deceased	1	1.4	1	2.6	0	0	
	cohabiting	3	4.3	3	7.7	0	0	
job	unemployed	21	30.0	8	20.5	13	41.9	0.14
	employed	42	60.0	26	66.7	16	51.6	
	underemployed	7	10.0	5	12.8	2	6.5	
HIV	+	10	14.3	5	12.8	5	16.1	0.47
	-	60	85.7	34	87.2	26	83.9	
current cigarette smoking	+	70	100	39	100	31	100	1.00
	-	0	0	0	0	0	0	
Risky alcohol use	+	19	27.1	11	28.2	8	25.8	0.52
	-	51	72.9	28	71.8	23	74.2	
past illicit drug use	+	68	97.1	37	94.9	31	100	0.31
	-	2	2.9	2	5.1	0	0	
attendance rate	100%	10	14.28	8	20.5	2	6.5	0.01 *
	90–99%	28	40	20	51.3	8	25.8	
	70–89%	17	24.29	6	15.4	11	35.5	
	less than 70%	15	21.4	5	12.8	10	32.3	

* *p* < 0.05.

**Table 2 ijerph-19-03703-t002:** Pretest in the intervention group and dropout group.

Variable	All Participants n = 70		Full Attendance Group n = 39		Dropout Group n = 31		*t* or (w)	*p*	Kolmogorov Smirnov (Test of Normality)		
	Mean (Rang)	SD	Mean (Rang)	SD	Mean (Rang)	SD			S	K	*p*
age	43.79 (34–71)	8.23	45.46 (35–66)	8.01	41.68 (34–71)	8.15	(−2.49)	0.013 *	0.86	0.18	0.014 *
onset age	25.49 (14–64)	9.33	27.21 (17–64)	9.61	23.32 (14–48)	8.63	(−2.17)	0.030 *	1.54	3.53	<0.001 ***
The average dose of methadone (mg/day)	38.18 (0–12)	9.33	41.28 (1–120)	24.96	34.29 (0–75.2)	20.13	1.27	0.210	0.87	1.62	0.200
1st TFAB	38.92 (12–64)	11.91	36.21 (12–60)	12.42	42.61 (20–64)	10.29	(−2.03)	0.043 *	−0.39	−0.67	0.007 **
1st DS-MV	39.73 (4–76)	14.35	36.49 (4–63)	14.19	43.81 (11–76)	13.69	−2.18	0.033 *	−0.27	−0.30	0.200
1st PHQ-9 total scores	9.86 (0–27)	6.86	8.62 (0–25)	6.97	11.42 (0–27)	6.50	−1.72	0.090	0.82	0.29	0.073
1st CS total scores	11.09 (0–29)	6.81	9.79 (0–29)	6.89	12.71 (0–21)	6.45	(−2.09)	0.037 *	0.72	−0.86	<0.001 ***

* *p* < 0.05; ** *p* < 0.01; *** *p* < 0.001; SD = standard deviation; S = skewness; K = kurtosis; w = Wilcoxon test.

**Table 3 ijerph-19-03703-t003:** Fully attended group repeated-measures ANOVA (n = 39).

	M	SD	Mauchly’ w	F	Effect Size (Partial *η^2^*)	Pairwise Comparisons
1st TFAB	36.21	12.4	0.951	7.662 **	0.172	1st TFAB = 2nd TFAB 2nd TFAB < 3rd TFAB * 1st TFAB < 3rd TFAB *
2nd TFAB	35.97	12.00				
3rd TFAB	39.92	10.94				
1st DS-MV	36.49	14.19	0.860	3.435 *	0.083	1st DS-MV = 2nd DS-MV 2nd DS-MV = 3rd DS-MV 1st DS-MV > 3rd DS-MV *
2nd DS-MV	32.69	17.06				
3rd DS-MV	29.72	15.13				
1st PHQ-9	8.62	6.97	0.725 **	^a^ 10.166 **	0.211	1st PHQ-9 > 2nd PHQ-9 * 2nd PHQ-9 > 3rd PHQ-9 * 1st PHQ-9 > 3rd PHQ-9 *
2nd PHQ-9	5.97	5.92				
3rd PHQ-9	4.28	4.26				
1st CS	9.79	6.89	0.777 **	^a^ 9.509 **	0.200	1st CS > 2nd CS * 2nd CS = 3rd CS 1st CS > 3rd CS *
2nd CS	6.13	7.52				
3rd CS	4.54	5.53				

* *p* < 0.05, ** *p* < 0.01. ^a^ Mauchly’s w test was used to test the homogeneity of variances. The Mauchly’s w values of the PHQ-9 and CS were significant. In addition, we used lower-bound corrections when the assumption of sphericity was violated.

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
