# Peer review of "Frontal Lobe Functions, Demoralization, Depression and Craving as Prognostic Factors and Positive Outcomes of Patients with Heroin Use Disorder Receiving 6 Months of Methadone Maintenance Treatment"

_ijerph, 2022, doi:10.3390/ijerph19063703_

Round 1

Reviewer 1 Report

Nowadays the definition of addiction as a chronic and relapsing brain disease has been worldwide accepted, and its deleterious consequences on physical health, psychology, and quality of life are widly known. Nevertheless, some recent surveys and studies have pointed out the impact of Covid19-pandemic on the epidemiology of Substance Use Disorders in general, highlighting the urgency of finding an effective management strategy for such health disorders. 

On this topic, this manuscript represents an interesting exploration on the direction of a patient-tailored therapy in Opioid Agonist Treatment: managing addiction, particularly Opioid Use Disorder (OUD), from a therapeutic viewpoint is definitely a challenging task for clinicians, as it implies extensive knowledge in managing long-term opioid medications and rehabilitation programmes.

Furthermore, the idea of exploring the frontal lobe functions appeared notable on this context. Studying dysfunctions in such brain circuitries represents nowadays a new research line on the background of mental disorders, aiming to deeper understand homogeneous symptom sets that cut across multiple disorders (ADHD, SUDs, etc). Deconstructing mental disorders in specific domains and focusing on these domains as new treatment targets that need distinct types of pharmacological treatments, researchers hopefully will further improve validity as well as treatment success rates.

The manuscript appears to be adequately clear and enough consistent in form and structure. The experimental design is appropriate to test the primary hypothesis, and results are reproducible. Conclusions are consistent with the evidence and the arguments presented.

Nevertheless, the accuracy of the manuscript appear to be partly affected by the underestimation of some relevant clinical aspects:

1- introduction might focus on how Methadone maintenance treatment is an effective intervention for OUD patients, as it allows a more effective reduction in heroin use than treatments that do not involve opioid medications [Mattick R. P., Breen C., Kimber J., Davoli M. (2009): Methadone Maintenance Therapy Versus No Opioid Replacement Therapy for Opioid Dependence. Cochrane Database Syst Rev(3): CD002209]. Nevertheless, studies have been conducted which showed also the significant effectiveness of such treatment on the psychopathological background of addiction [Mohammadi M, Kazeminia M, Abdoli N, Khaledipaveh B, Shohaimi S, Salari N, Hosseinian-Far M. The effect of methadone on depression among addicts: a systematic review and meta-analysis. Health Qual Life Outcomes. 2020 Nov 23;18(1):373. doi: 10.1186/s12955-020-01599-3. PMID: 33225933; PMCID: PMC7681984][Cao P, Zhang Z, Zhong J, Xu S, Huang Q, Fan N. Effects of treatment status and life quality on anxiety in MMT patients. Subst Abuse Treat Prev Policy. 2021 Jan 13;16(1):9. doi: 10.1186/s13011-021-00343-4. PMID: 33441176; PMCID: PMC7805232].

2- Some sentences miss the conclusion (i.e. final lines of abstract).

3- High relevance has been assigned to 'demoralization' items, and the take-home point is focused on the relationship between such psychic background as prognostic factors and positive outcomes of treatment. Nevertheless, the clinical difference between 'demoralization' and depression is not well described. What's the theshold of demoralization? How a physician can clinically distinguish whether a patient is demoralised or depressed Characterizing those clinical features may improve the quality of the presentation.

Author Response

Please see the attachment (p1~p5)

Reviewer 2 Report

The study has a major flaw which calls into question the predictions being made.  The problem is that the sample size is too small to generate statistical power.  Statistical power is need to draw accurate conclusions about a population using sample data.  A sample of 39 participants is not enough for statistical power and drawing conclusions about the population who remained in treatment.  The two "clinical implications" cannot be supported by the study.

Author Response

Please see the attachment (p1~2, and p5)

Reviewer 3 Report

Thank you for the opportunity to review this manuscript.

The quality of the writing facilitates easy reading and understanding.

I have some major comments.

- The authors should better describe the program of care provided in addition to the prescription and administration of MMT.

- Are serotonin reuptake inhibitors introduced?

- How tobacco and alcohol use are changing?

- In the method and in the discussion, the authors do not clearly distinguish the difference between the pharmacological effect of the treatment and the effect of stabilization and global care. Therapy is not limited to the administration of methadone during a MMT protocol.

What psychotherapies are used?

The focus on the Brain frontal lobe function is laudable. But, is there cognitive remediation?

I have also some minor comments.

In Introduction

- In the introduction, several factors influencing the effectiveness of methadone maintenance treatment (MMT) are mentioned, including the duration of treatment (l. 42). The personalization of the dose is also important, as you say in the discussion, with many factors which affect the level of efficient dose (Mouly et al. DOI: 10.1111/bcp.12576). This is interesting to mention in introduction because the mean dose of the study’s participants is low and the dispersion is large (table 2).

In Materials and Methods

- For the “Diagnostic and Statistical Manual of Mental Disorders, Fifth Edition”, it is “DSM-5”, not “DSM-V” (l. 83). This is confusing with the demoralization scale (DS-MV).

- Regarding the criteria, these are not “exclusion criteria” but non-inclusion criteria (l. 85).

- The non-inclusion criterion “under 18 years old or over 65 years old” is superfluous (l. 85). The inclusion criterion “aged 18 to 65 years old” is sufficient.

- There is no control group to respond to the primary question.

- There is no control of abstinence.

In Results

- What do you mean by “drinkers” (l. 148) and “drinking” (table 1)? This suggests that it is alcohol use, but it is required to specify it. Furthermore, what is the criterion used for this qualification? What is the frequency of alcohol use? Are there any DSM-V criteria?

- Similarly, regarding the “smoking” (table 1 and l. 148), what is the categorization used? And it is necessary to add the term “tobacco”.

- Cannabis, benzodiazepine, GHB or cocaine uses are not criteria for non-inclusion. What are the “other substances”?

The notable absence of other substances use makes this a very special population.

- In table 2, does the methadone dose have a normal distribution? This is important because it is essential for the use of the t test of Student. And the standard deviations are not similar.

The number of participants is just over 30 (n=31 in “dropout group”), limit admitted to exempt from the condition of normality.

But, using a more appropriate test may show a different result. That a non-parametric test shows? Is there a concordance?

If is not a normal distribution, is it a quasi-poisson distribution?

- I have the same interrogation with “1st PHQ-9 total scores” and “1st CS total scores”.

- The majority of the p-values are less than 0.10. There may be a lack of power. This is not sufficient to conclude that there are significant differences, but it does suggest that the results in the “Full attendance group” group should be taken with caution. The main result concern one particular subgroup. It is nonetheless relevant.

- Line 174, the use of “DSMV” for the “demoralization scale” is confusing, with the “DSM-V” (l 83). Line 93, the term used is “DS-MV”.

- Why a correction for multiple testing was not used in the pairwise comparisons? For example, the False Discovery Rate.

- The paragraph “but the 2nd TFAB (M = 32.69, SD = 17.06) and 3rd TFAB (p < .176) showed no differences; there was no difference in average total intake between the 1st TFAB and 2nd TFAB (p =.213). This indicates that demoralization (TFAB scale score) significantly improved following continuous treatment for six months (see Table 3).” (lines 177-180) needs to be corrected. There is an error between TFAB and DS-MV.

In Discussion

- The first sentence permutes the answers to the questions 1 and 2.

For reminder; (lines 67-70)

“the purpose of this study was:

    1. To explore the severity of demoralization in patients receiving MMT.
    2. To explore demoralization or frontal lobe function as prognostic factors to predict retention rate.”

- The paragraph limits must be revised.

Author Response

Please see the attachment (p1~2, and p5~14)

Round 2

Reviewer 3 Report

Thank you for your answers.

However, there is an inaccuracy in the results that you have not corrected. In the paragraph of lines 185 – 192 (p. 6), you wrote « TFAB » at the place of « DS-MV » (example « demoralization (TFAB scale score) » (line 191). Please make the correction.

Author Response

Dear Editor and Reviewers,

Thank you for your letter dated 4th March, 2022. We thank the reviewers for the time and effort that they have put into reviewing the previous version of the manuscript. Based on the instructions provided in your letter, we uploaded the file of the revised manuscript. Accordingly, we have uploaded a copy of the original manuscript with all the changes highlighted by using the track changes mode in MS Word. The comments are reproduced and our responses are given directly afterward in a different color. We would like also to thank you for allowing us to resubmit a revised copy of the manuscript.

We hope that the revised manuscript is accepted for publication in the International Journal of Environmental Research and Public Health.

Sincerely,

Chang, Shan-Mei

2022.03.04

#Reviewer 3

Suggestions for Authors

Thank you for your answers.

However, there is an inaccuracy in the results that you have not corrected. In the paragraph of lines 185 – 192 (p. 6), you wrote « TFAB » at the place of « DS-MV » (example « demoralization (TFAB scale score) » (line 191). Please make the correction.

Response:

Thanks of suggestion, the manuscript had been edited. Modified throughout the text according to the comment:

「This indicated that frontal lobe function (TFAB scale score) significantly improved following continuous treatment for six months」
